# Copper-Promoted Cross-Coupling Reactions for the Synthesis of Aryl(difluoromethyl)phosphonates Using Trimethylsilyl(difluoromethyl)phosphonate

**DOI:** 10.3390/molecules23123292

**Published:** 2018-12-11

**Authors:** Kazuki Komoda, Rei Iwamoto, Masakazu Kasumi, Hideki Amii

**Affiliations:** 1Division of Molecular Science, Graduate School of Science and Technology, 1-5-1, Tenjin-cho, Kiryu, Gunma 376-8515, Japan; t162a004@gunma-u.ac.jp (K.K.); t11801125@gunma-u.ac.jp (M.K.); 2Department of Chemistry, Graduate School of Science, Kobe University, Nada-ku, Kobe 657-8501, Japan; qq5f25s9n@energy.ocn.ne.jp

**Keywords:** phosphonates, difluoromethyl, cross-coupling, copper, silicon

## Abstract

A convenient and effective route for the synthesis of aryl(difluoromethyl)phosphonates has been developed based on cross-coupling reactions. Upon treatment with a stoichiometric amount (or a catalytic amount in some cases) of CuI and CsF, aryl iodides reacted smoothly with (silyldifluoromethyl)phosphonates to give the corresponding aryl(difluoromethyl)phosphonates in good yields.

## 1. Introduction

Fluorinated organic compounds play important roles in the progress of medicinal, agricultural, and material sciences [1,2,3,4,5,6,7,8,9]. Difluoromethylene compounds have been important synthetic targets due to the unique properties of the CF_2_-moiety, which acts as a bioisostere for an ether oxygen atom or a carbonyl group [10,11,12]. Organic compounds containing difluoromethylphosphonate moieties (-CF_2_PO(OR)_2_) are of great interest for application as protein tyrosine phosphatase (PTP) inhibitors [13,14,15,16,17]. Among them, aryl(difluoromethyl)phosphonates (Ar-CF_2_PO(OR)_2_) have received a great deal of attention in the design and development of bioactive agents. Currently, the selective introduction of CF_2_PO(OR)_2_ groups into aromatic compounds is a topic of considerable interest. Meanwhile, transition metal-catalyzed cross-coupling reactions are now employed for wide repertoire of aromatic carbon-carbon, carbon-heteroatom bond-forming reactions [18]. Fluoro and fluoroalkyl cross-coupling reactions are one of the most powerful methods to construct fluoro aromatic compounds [19,20,21]. To date, selective introduction of difluoromethylphosphonate groups into aromatics with high generality and functional-group compatibility has been investigated broadly [22,23,24,25,26,27,28,29,30,31,32,33,34,35,36]. For the construction of Ar-CF_2_PO(OR)_2_ frameworks, one of the most common and reliable methods is copper-mediated cross-coupling reactions involving CuCF_2_PO(OR)_2_ species [22,23,25,26,27,28,32,33,35]. Despite such synthetic utility, most of the reported methods require the use of stoichiometric (sometimes, excess) amounts of copper reagents to complete these cross-coupling reactions [26,28]. There have been few and limited successful examples in which a small amount of transition metal complexes promoted cross-coupling reactions for introducing difluoromethyl-phosphonate moieties. Due to the demand for Ar-CF_2_PO(OR)_2_ compounds, a general catalytic approach is required. As a cross-coupling participant, [(trimethylsilyl)difluoromethyl]phosphonate (Me_3_Si-CF_2_PO(OEt)_2_: **2**) is stable and readily available from Br-CF_2_PO(OEt)_2_ [37]. Herein, we describe fundamental studies on Cu-mediated cross-coupling of aryl iodides **1** and [(silyl)difluoromethyl]phosphonate **2** to deliver Ar-CF_2_PO(OEt)_2_
**3** (Scheme 1).

## 2. Results and Discussion

Initially, we surveyed the suitable reaction conditions for Cu-promoted cross-coupling of aryl iodides **1** with (silyldifluoromethyl)phosphonates **2**. When a mixture of 4-iodobenzonitrile (**1a**) and [(trimethylsilyl)difluoromethyl]phosphonates (**2**) in toluene was heated at 60 °C for 24 h in the presence of CuI (1.0 equiv.) and KF (1.2 equiv.), the desired cross-coupling product **3a** was not obtained at all (Table 1, Entry 1). Next, we conducted the reaction using a polar aprotic solvent such as DMSO, and as a consequence, aryl(difluoromethyl)phosphonate **3a** was obtained in 43% NMR yield (Entry 2). The use of amide solvents such as NMP and DMF was effective for the formation of phosphonate **3a** in 55% and 76% yields, respectively (Entries 3 and 4). Furthermore, THF was one of the promising solvents for the Cu-mediated transformation to afford **3a** in 74% NMR yield (Entry 5). Then, we focused our attention on fluoride sources. Instead of potassium fluoride, the use of tetrabutylammonium fluoride (TBAF) in DMF resulted in protodesilylation of **2** to yield HCF_2_PO(OEt)_2_ as a major product (Entry 6). Cesium fluoride was found to be effective to the cross-coupling reaction Entry 7). The combination of CsF as a fluoride source and THF as a solvent gave the highest product yield of **3a** (Entry 8).

With optimized reaction conditions in hand, other examples of the selective formation of aryl(difluoromethyl)phosphonate **3** were tested (Table 2). Upon treatment with a stoichiometric amount of CuI, a wide repertoire of aryl iodides **1** underwent cross-coupling reactions to give the corresponding difluoromethylphosphonates **3** in moderate to good yields. Of significant interest, the present protocol worked well for both electron-deficient and electron-rich iodoarenes (**1a**–**d**). Cyano and ester groups in **1a** and **1b** were tolerable under the nucleophilic fluoroalkylating conditions. The cross-coupling of 1-iodonaphthalene (**1e**) with silyl phosphonate **2** proceeded to afford naphthyl difluoromethylphosphonate **3e**. Furthermore, heteroarenes **1f** and **1g** underwent the difluoromethylenephosphonation to give **3f** and **3g** in 51% and 70% isolated yields, respectively. Notably, chloro and bromo functionalities in **3h** and **3i** were compatible with the present transformation. In the each cross-coupling of **1** with **2**, the major by-product was HCF_2_PO(OEt)_2_; for instance, the reaction of 2-iodopyridine (**1f**) with **2** gave 25% of HCF_2_PO(OEt)_2_ besides the desired Py-CF_2_P(O)(OEt)_2_ (**3f**). With a good level of functional group tolerance, the reactions proceeded smoothly under mild conditions.

Copper-mediated cross-coupling of aryl iodides with BrZnCF_2_PO(OEt)_2_ is one of the most reliable methods for the synthesis of Ar-CF_2_PO(OEt)_2_ species (Scheme 2) [23]. However, there is a technical drawback for the transmetallation methodology using BrZnCF_2_PO(OEt)_2_ and BrCdCF_2_PO(OEt)_2_; in most cases, the use of stoichiometric amounts of copper salts is needed to complete the cross-coupling reactions. Poisson and co-workers reported the cross-coupling reactions of Me_3_Si-CF_2_PO(OEt)_2_ (**2**) with aryl diazonium salts and iodonium salts upon exposure to a stoichiometric amount of CuSCN [35]. In 2012, Zhang overcame the shortcoming of Cu-mediated cross-coupling for install of CF_2_PO(OEt)_2_ group into aryl rings [26]. As a key strategy, employing the aryl iodides (or aryl bromides) possessing directing substituents such as methoxycarbonyl or triazene groups at the ortho-position in aryl iodides to facilitate the oxidative addition. Anyway, in order to accomplish the reactions promoted by a small amount of copper complexes, there is a limitation concerning the substrates endowed with ortho-directing groups [26,28]. 

We examined the reactions of aryl iodides **1** with (silyldifluoromethyl)phosphonate **2** by the use of a small amount of CuI (Scheme 3). 

When a mixture of **1a**, phosphonates **2** and CsF in THF was heated at 60 °C for 24 h in the presence of CuI (0.5 equiv. to **1a**) under argon atmosphere, the cross-coupling reaction proceeded to give **3a** in 71% yield. Upon treatment with further reduced amount of CuI (0.2 equiv. to **1a** and **1f**), the cross-coupling products **3a** and **3f** were obtained in 50% and 42% NMR yields, respectively. Interestingly, iodoquinoline participated in cross-coupling reaction to afford difluoromethylphosphonate **3g** in 69% yield at 10 mol% catalyst loading. Control of the rate of the generation of CF_2_PO(OEt)_2_ anion by the reaction of Me_3_Si-CF_2_PO(OEt)_2_ (**2**) with CsF would render a reaction catalytic in copper possible. The major role of CsF is the activation of organosilicon compound **2** to generate CF_2_PO(OEt)_2_ anion. The low solubility of CsF in THF would contribute to slow generation of CF_2_PO(OEt)_2_ anion and formation of CuCF_2_PO(OEt)_2_ species in the catalytic reactions. 

## 3. Materials and Methods

### 3.1. General Information

All reactions were carried out under an argon atmosphere in flame-dried glassware. Syringes used to transfer anhydrous solvents or reagents were purged with argon prior to use. Most chemicals were purchased from commercial suppliers and used without further purification. THF was dried by reflux over Na chips in the presence of benzophenone as indicator. Analytical TLC was performed on aluminum silica gel 60 F_254_ (Merck, Darmstadt, Germany) sheets, which were visualized by the quenching of UV fluorescence (254 nm). Column chromatography was conducted on silica gel (Cica, 60–210 mesh, spherical, neutral). NMR spectra were acquired on a JNMECS 400 (400 MHz for ^1^H and 376 MHz for ^19^F, respectively) spectrometer (JEOL, Ltd., Tokyo, Japan). ^1^H-NMR spectra were recorded using TMS (Me_4_Si) as internal standard (δ = 0). ^19^F-NMR spectra were recorded using hexafluorobenzene (C_6_F_6_) as internal standard (δ = 0). All the ^1^H-NMR and ^19^F-NMR spectra of **3** matched those reported previously [23,25,31,35].

### 3.2. Cross-Coupling of Aryl Iodides with Me_3_SiCF_2_PO(OEt)_2_

To a mixture of CuI (95.2 mg, 0.50 mmol), CsF (91.1 mg, 0.60 mmol), 4-iodobenzonitrile (**1a**, 114.0 mg, 0.50 mmol), and THF (1.0 mL) was added [(trimethylsilyl)difluoromethyl]phosphonate (**2**, 156.2 mg, 0.60 mmol) at room temperature. The reaction mixture was stirred at 60 °C in an atmosphere of nitrogen for 24 h and quenched with water. The aqueous layer was extracted with ethyl acetate. Then, the combined organic phase was washed with water, dried over anhydrous Na_2_SO_4_. The crude product was purified by column chromatography on silica gel (hexane/EtOAc = 2/1) to give **3a** (121.4 mg, 0.42 mmol, 84%) as a pale-yellow oil.

*Diethyl (4*-*cyanophenyl)difluoromethylphosphonate* (**3a**). ^1^H-NMR (CDCl_3_): 7.82–7.71 (m, 4H), 4.35–4.18 (m, 4H), 1.34 (t, *J* = 7.2 Hz, 6H); ^19^F-NMR (CDCl_3_): 51.7 (d, *J* = 112.8 Hz, 2F).

*Ethyl 4*-*[(diethoxyphosphoryl)difluoromethyl]benzoate* (**3b**). Following a general procedure, CuI (95.2 mg, 0.50 mmol), CsF (91.1 mg, 0.60 mmol), ethyl 4-iodobenzoate (**1b**, 138.0 mg, 0.50 mmol), [(trimethylsilyl)difluoromethyl]phosphonate (**2**, 156.2 mg, 0.60 mmol) were used in THF (1.0 mL) at 60 °C for 24 h. The crude product was purified by column chromatography on silica gel (hexane/EtOAc = 2/1) to give product **2b** (77.4 mg, 46% yield) as a pale yellow oil. ^1^H-NMR (CDCl_3_): 8.13 (2H, d, *J* = 8.4), 7.70 (2H, d, *J* = 8.4), 4.41 (2H, q, *J* = 7.2), 4.26–4.12 (4H, m), 1.41 (3H, t, *J* = 7.2), 1.32 (6H, t, *J* = 7.2); ^19^F-NMR (CDCl_3_): 52.4 (d, *J* = 112.8 Hz, 2F).

*Diethyl difluoro(phenyl)methylphosphonate* (**3c**). Following a general procedure, CuI (95.2 mg, 0.50 mmol), CsF (91.1 mg, 0.60 mmol), iodobenzene (**1c**, 102.0 mg, 0.50 mmol), [(trimethylsilyl)-difluoromethyl]phosphonate (**2**, 156.2 mg, 0.60 mmol) were used in THF (1.0 mL) at 60 °C for 24 h. The crude product was purified by column chromatography on silica gel (hexane/EtOAc = 3/1) to give product **2c** (76.1 mg, 57% yield) as a pale-yellow oil. ^1^H-NMR (CDCl_3_): 7.63 (2H, d, *J* = 8.0), 7.50–7.45 (3H, m), 4.26–4.13 (4H, m), 1.31 (6H, t, *J* = 7.0); ^19^F-NMR (CDCl_3_): 53.2 (d, *J* = 116.6 Hz, 2F). 

*Diethyl (4*-*methoxyphenyl)difluoromethylphosphonate* (3d) Following a general procedure, CuI (95.2 mg, 0.50 mmol), CsF (91.1 mg, 0.60 mmol), 1-ethoxy-4-iodobenzene (**1d**, 117.0 mg, 0.50 mmol), [(trimethylsilyl)difluoromethyl]phosphonate (**2**, 157.5 mg, 0.61 mmol) were used in THF (1.0 mL) at 60 °C for 24 h. The crude product was purified by column chromatography on silica gel (hexane/EtOAc = 3/1) to give product **2d** (51.1 mg, 36% yield) as a colorless oil. ^1^H-NMR (CDCl_3_,): 7.95 (2H, d, *J* = 7.6), 7.54 (2H, d, *J* = 7.6), 4.23–4.10 (4H, m), 3.83 (3H, s), 1.30 (6H, t, *J* = 6.8); ^19^F-NMR (CDCl_3_): 54.6 (d, *J* = 115.8 Hz, 2F).

*Diethyl difluoro(naphthalen-1-yl)methylphosphonate* (**3e**). Following a general procedure, CuI (95.2 mg, 0.50 mmol), CsF (91.1 mg, 0.60 mmol), 1-iodonaphthalene (**1e**, 127.0 mg, 0.50 mmol), [(trimethylsilyl)difluoromethyl]phosphonate (**2**, 156.2 mg, 0.60 mmol) were used in THF (1.0 mL) at 60 °C for 24 h. The crude product was purified by column chromatography on silica gel (hexane/EtOAc = 3/1) to give product **2e** (77.1 mg, 49% yield) as a colorless oil. ^1^H-NMR (CDCl_3_): 8.44 (1H, d, *J* = 8.4), 7.97 (1H, d, *J* = 8.4), 7.88 (1H, d, *J* = 8.4), 7.81 (1H, d, *J* = 7.2), 7.60–7.48 (3H, m), 4.24–4.00 (4H, m), 1.27 (6H, t, *J* = 7.2); ^19^F-NMR (CDCl_3_): 53.2 (d, *J* = 112.8 Hz, 2F).

*Diethyl difluoro(pyridin-2-yl)methylphosphonate* (**3f**). Following a general procedure, CuI (95.2 mg, 0.50 mmol), CsF (91.1 mg, 0.60 mmol), 2-iodopyridine (**1f**, 102.5 mg, 0.50 mmol), [(trimethylsilyl)difluoromethyl]phosphonate (**2**, 157.6 mg, 0.61 mmol) were used in THF (1.0 mL) at 60 °C for 24 h. The crude product was purified by column chromatography on silica gel (hexane/EtOAc = 2/1) to give product **2e** (67.0 mg, 51% yield) as a pale yellow oil. ^1^H-NMR (CDCl_3_): 8.72 (1H, d, *J* = 4.4), 7.85 (1H, t, *J* = 8.0), 7.71 (1H, d, *J* = 8.0), 7.44–7.41 (1H, m), 4.24–4.37 (4H, m), 1.36 (6H, t, *J* = 7.0); ^19^F-NMR (CDCl_3_): 51.1 (d, *J* = 109.8 Hz, 2F).

*Diethyl difluoro(quinolin-2-yl)methylphosphonate* (**3g**). To a mixture of CuI (9.5 mg, 0.05 mmol), CsF (91.1 mg, 0.60 mmol), 2-iodoquinoline (**1g**, 127.5 mg, 0.50 mmol), [(trimethylsilyl)difluoromethyl]-phosphonate (**2**, 151.0 mg, 0.58 mmol), and THF (1.0 mL) was added [(trimethylsilyl)-difluoromethyl]phosphonate (**2**, 156.2 mg, 0.60 mmol) at room temperature. The reaction mixture was stirred at 60 °C in an atmosphere of nitrogen for 24 h and quenched with water. The aqueous layer was extracted with ethyl acetate. Then, the combined organic phase was washed with water, dried over anhydrous Na_2_SO_4_. The crude product was purified by column chromatography on silica gel (hexane/EtOAc = 2/1) to give **3g** (110.0 mg, 69% yield) as a pale yellow oil. ^1^H-NMR (CDCl_3_,): 8.32 (1H, d, *J* = 8.4), 8.19 (1H, d, *J* = 8.4), 7.89 (1H, d, *J* = 8.0), 7.81–7.77 (1H, m), 7.64 (1H, t, *J* = 8.0), 4.39–4.32 (4H, m), 1.38 (6H, t, *J* = 7.2); ^19^F-NMR (CDCl_3_): 51.1 (d, *J* = 103.8 Hz, 2F).

*Diethyl (3,4-dichlorophenyl)difluoromethylphosphonate* (**3****h**). Following a general procedure, CuI (91.4 mg, 0.48 mmol), CsF (89.3 mg, 0.59 mmol), 3,4-dichloro-1-iodobenzene (**1h**, 131.7 mg, 0.48 mmol), [(trimethylsilyl)difluoromethyl]phosphonate (**2**, 153.1 mg, 0.58 mmol) were used in THF (1.0 mL) at 60 °C for 24 h. The crude product was purified by column chromatography on silica gel (hexane/EtOAc = 3/1) to give product **3h** (71.2 mg, 44% yield) as a pale yellow oil. ^1^H-NMR (CDCl_3_): 7.70 (1H, s), 7.55 (1H, d, *J* = 8.3), 7.47 (1H, d, *J* = 8.3), 4.33–4.14 (4H, m), 1.35 (6H, t, *J* = 7.0); ^19^F-NMR (CDCl_3_): 52.8 (2F, d, *J* = 113.4 Hz).

*Diethyl (4-bromophenyl)difluoromethylphosphonate* (**3i**). Following a general procedure, CuI (95.2 mg, 0.50 mmol), CsF (91.1 mg, 0.60 mmol), 4-bromo-1-iodobenzene (**1i**, 142.0 mg, 0.50 mmol), [(trimethylsilyl)difluoromethyl]phosphonate (**2**, 156.2 mg, 0.60 mmol) were used in THF (1.0 mL) at 60 °C for 24 h. The crude product was purified by column chromatography on silica gel (hexane/EtOAc = 3/1) to give product **3i** (100.0 mg, 58% yield) as a pale yellow oil. ^1^H-NMR (CDCl_3_): 7.60 (2H, d, *J* = 8.4), 7.49 (2H, d, *J* = 8.4), 4.10–4.30 (4H, m), 1.33 (6H, t, *J* = 7.1); ^19^F-NMR (CDCl_3_): 52.8 (2F, d, *J* = 113.4 Hz).

## 4. Conclusions

In summary, we have developed a convenient route to aryl(difluoromethyl)phosphonates from aryl iodides. Using a simple combination of the coupling partners (iodoarenes and Me_3_Si-CF_2_PO(OEt)_2_), the cross-coupling proceeded smoothly under mild reaction conditions. The present transformations employing CuI are synthetically useful. In some cases, a small amount of CuI promoted the cross-coupling reactions to afford aryl(difluoromethyl)phosphonates. From the practical viewpoint, this study will enable the development of valuable organofluorine compounds with potential biological utility.

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
