# Peer review of "Copper-Promoted Cross-Coupling Reactions for the Synthesis of Aryl(difluoromethyl)phosphonates Using Trimethylsilyl(difluoromethyl)phosphonate"

_molecules, 2018, doi:10.3390/molecules23123292_

Round 1

Reviewer 1 Report

These results will be helpful to the scientific community and publishable. The introduction is too brief lacking the applications of such phosphonates in different fields such as material chemistry, biochemistry and different phosphonate based catalysis. So, my suggestion that the introduction part could be slightly expanded.

Author Response

Thank you for the useful suggestion. In this brief paper, we focused on the synthetic methodology. As pointed out, aryl difluoromethylphosphonates are important bioactive compounds. Actually, many research groups are investigating the synthetic routes to aryl difluoromethylphosphonates. As an introductory backgrounds, we divided into two parts (“stoichiometric in copper” in Introduction and “catalytic in copper” in Results and Discussion). 

Reviewer 2 Report

The ms of Amii et al. is on the Cu-promoted cross-coupling reactions of fluorinated compounds.

The topic is interesting, however the ms suffers some major limitations.

First of all, the ms is not well written and should be improved in general (concerning structure and English usage as well).

- In the Introduction, references [19-21], [29-31] and [34] are missing or not cited.(Later on ref [36] isn't cited either.) The order of references is also mixed up. 

- On page 3, lines 85-97 contain literature background, therefore this part should be moved from the R&D to the Introduction section.

- Line 74: "the cross-coupling participants 2 are stable and readily available..." - this statement should be mentioned earlier.

Another major drawback is that the Referee doesn't see any major improvements (or novelty) as compared to the literature methods.

In the Introduction, authors claim that "most of the reported examples required the use of stoichiometric amounts of copper reagent" - however, the submitted ms describes a protocol applying also a stoichiometric amount of CuI (6 examples), with only 2 examples for a reduced amount of Cu reagent.

Moreover, none of the compounds synthesized are new, and the yields are even lower than that of the literature examples (see for eg. 3b: 46% as compared to 84% [23], 3c: 57% vs 80%[23] or 3d: 36% vs 85% [23]).

The method is not completely new either. It should be improved and extended for new compounds. 

The role of CsF should be discussed.

Why the preparative yield is low in general? What about the formation of HCF2PO(OEt)2 as a potential by-product?

What does "semi-catalytic ampint" mean?

What is the advantage of using Me3SiCF2PO(OEt)2 instead of (for eg.) BrZNCF2PO(OEt)2 when 1 equiv. of CuI is applied? Me3SiCF2PO(OEt)2 was already applied in the literature in Pd-promoted C-C couplings of aryl iodides and only moderate yields could be obtained as in this case as well.

Other problems:

 English usage is poor in some cases. A few examples are:

- "reported examples are required the use of..."

- "formation phosphonate 3a"

- "each yields was"

- "1f" partook in the.."

etc.

NMR data should be assigned and systematically compared with that of the literature data instead of saying "spectra matched that reported by XY".

In conclusion, the ms cannot be accepted in it's current form but may be reconsidered after facing the above mentioned problems. 

Author Response

- In the Introduction, references [19-21], [29-31] and [34] are missing or not cited. (Later on ref [36] isn't cited either.) The order of references is also mixed up.

As pointed out, citation of references [29-31] is missing in the main text. We have added the citation of references [29-31] as follows.
Page 1, lines 28-30; Fluoro and fluoroalkyl cross-coupling reactions are one of the most powerful methods to construct fluoro aromatic compounds [19-21].

Citation of references [29-31] and [34] was already done.

Page 1, lines 30-32; To date, selective introduction of difluoromethylphosphonate groups into aromatics with high generality and functional-group compatibility has been investigated broadly [22-36].

- On page 3, lines 85-97 contain literature background, therefore this part should be moved from the R&D to the Introduction section.

Mainly, we described Cu-mediated cross-coupling (stoichiometric in copper) in this paper. We divided into two parts (“stoichiometric in copper” and “catalytic in copper”). Not only our results, but the previous reports by other groups have certain limitations. In the later R&D section (“catalytic in copper”), we want to show literature background.

- Line 74: "the cross-coupling participants 2 are stable and readily available..." - this statement should be mentioned earlier.

We have moved the sentence to the Introduction section and have changed partially.

Page 1, lines 39-42; As a cross-coupling participant, [(trimethylsilyl)difluoromethyl]phosphonate (Me3Si-CF2PO(OEt)2: 2) is stable and readily available from Br-CF2PO(OEt)2 [37]. Herein, we describe fundamental studies on Cu-mediated cross-coupling of aryl iodides 1 and [(silyl)difluoromethyl]phosphonate 2 to deliver Ar-CF2PO(OEt)2 3 (Scheme 1).

Another major drawback is that the Referee doesn't see any major improvements (or novelty) as compared to the literature methods.
In the Introduction, authors claim that "most of the reported examples required the use of stoichiometric amounts of copper reagent" - however, the submitted ms describes a protocol applying also a stoichiometric amount of CuI (6 examples), with only 2 examples for a reduced amount of Cu reagent.

Moreover, none of the compounds synthesized are new, and the yields are even lower than that of the literature examples (see for eg. 3b: 46% as compared to 84% [23], 3c: 57% vs 80%[23] or 3d: 36% vs 85% [23]).
The method is not completely new either. It should be improved and extended for new compounds.

We described the simple combination involving nucleophilic difluoromethylating reagents and aryl iodides for cross-coupling. It seems an orthodox approach, however even now, there have been many publications on Cu-mediated aromatic fluoroalkylation which need the use of a stoichiometric amount of copper salts. In this paper, we would like to disclose a new entry of Cu-catalyzed difluoromethylation.

Furthermore, in Table 2, we have added two examples (3h and 3i) with chloro- and bromo-functionalities. 

The role of CsF should be discussed.

The major role of CsF is the activation of Me3Si-CF2PO(OEt)2 to generate CF2PO(OEt)2 anion. However, the real role (the behavior in THF solvent) is not clear due to the low solubility of CsF in organic solvents such as ethers.

Why the preparative yield is low in general? What about the formation of HCF2PO(OEt)2 as a potential by-product?

The major by-product was HCF2PO(OEt)2. For instance, the reaction of Py-I (1f) with 2 gave 25% of HCF2PO(OEt)2 besides the desired Py-CF2P(O)(OEt)2 (3f).

What does "semi-catalytic ampint" mean?

We have changed the sentence as follows (with additional examples).

We examined the reactions of aryl iodides 1 with(silyldifluoromethyl)phosphonate 2 by the use of a small amount of CuI (Scheme 3). When a mixture of 1a, phosphonates 2 and CsF in THF was heated at 60 °C for 24 h in the presence of CuI (0.5 equiv to 1a) under argon atmosphere, the cross-coupling reaction proceeded to give 3a in 71% yield. Upon treatment with further reduced amount of CuI (0.2 equiv to 1a and 1f), the cross-coupling products 3a and 3f were obtained in 50% and 42% NMR yields, respectively.

What is the advantage of using Me3SiCF2PO(OEt)2 instead of (for eg.) BrZNCF2PO(OEt)2 when 1 equiv. of CuI is applied? Me3SiCF2PO(OEt)2 was already applied in the literature in Pd-promoted C-C couplings of aryl iodides and only moderate yields could be obtained as in this case as well.

We focus on copper catalysts due to their abundance and inexpensiveness.

Before using Me3SiCF2P(O)(OEt)2, we conducted many experiments using BrZnCF2PO(OEt)2 in the presence of a small amount of Cu salts. However, TON of the Cu-catalyzed cross-coupling of simple aryl iodides was low (TON: 1.0-1.2). The present studies (the use of Me3SiCF2P(O)(OEt)2) would provide a synthetic potential of Cu- catalyzed transformations.

Other problems:
English usage is poor in some cases. A few examples are:
- "reported examples are required the use of..."
We have changed as follows.
Despite such synthetic utility, most of the reported methods require the use of stoichiometric (sometimes, excess) amounts of copper reagents

- "formation phosphonate 3a"We have changed as follows.

formation of phosphonate 3a

- "each yields was"

We have changed as follows (the footnote of Table 1).

Each yield was calculated by 19F NMR analysis

- "1f" partook in the.."
We have changed as follows with another example.
Furthermore, heteroarenes 1f and 1g underwent the difluoromethylenephosphonation to give 3f and 3g in 51% and 70% isolated yields, respectively.

NMR data should be assigned and systematically compared with that of the literature data instead of saying "spectra matched that reported by XY".

We have removed the corresponding sentences, moves into “the former part of 3. Materials and Methods” and changed as follows.
All the 1H NMR and 19F NMR spectra of 3 matched those reported by previous reports [23, 25, 31]. 

Reviewer 3 Report

The paper by Hideki Amii et al. describes improvements in experimental conditions of a well known reaction of aryl iodides with trimethylsilyl(difluoromethyl)phosphonate. The authors achieved significantly higher yields of the products in comparison to previous reports. However, for a paper designed for optimization of reaction conditions I would expect a little bit more systematic studies:

1.      Most significant experiments (Table 1, entries 3,4,5,7,8) should be repeated to confirm the results.

2.      Reactions shown in Scheme 3 should be performed with the same amounts of CuI to be comparable.

3.      For the model reaction of 1a + 2 more concentrations of CuI should be examined, e.g., 0.5, 0.1 and 0.05 eq. The possibility of using sub-stoichiometric amounts of CuI are presented in the manuscript as an important achievement; however it was studied very superficially.

4.      Compound 1g should be included in studies shown in Table 2. Otherwise it is not possible to conclude the influence of the amount of CuI on the yield in this case.

Minor corrections required:
Line 56: “highest” instead of “higher”

Scheme 2, reaction 3: a typo in oC under the arrow
Scheme 3: 19F NMR yields or isolated yields are shown?

Lines 116 & 117: something is missing in “(     = 0)”

Pages 5 & 6: symbols 3a–3g should be bolded.

Author Response

1. Most significant experiments (Table 1, entries 3,4,5,7,8) should be repeated to confirm the results.

Initially, we conducted the cross-coupling reactions in DMF (Entry 4) and obtained 3a-3g in similar chemical yields to those using THF solvent. Major drawback using DMF as a solvent is the difficulty to perform the reaction catalytic in copper.

2. Reactions shown in Scheme 3 should be performed with the same amounts of CuI to be comparable.
3. For the model reaction of 1a + 2 more concentrations of CuI should be examined,

e.g., 0.5, 0.1 and 0.05 eq. The possibility of using substoichiometric amounts of CuI are presented in the manuscript as an important achievement; however it was studied very superficially.

We have completely changed Scheme 3 involving the results of the different concentrations of CuI. 

4. Compound 1g should be included in studies shown in Table 2.
Otherwise it is not possible to conclude the influence of the amount of CuI on the yield in this case.

In Table 2, we have included the result employing compound 1g with a stoichiometric amount of CuI. Furthermore, in Table 2, we have added two examples (3h and 3i) with chloro- and bromo-functionalities. 

Minor corrections required:
Line 56: “highest” instead of “higher”
We have changed as follows.
... gave the highest product yield of 3a (Entry 8).

Scheme 2, reaction 3: a typo in oC under the arrow

It would be a corrupted letter.

Scheme 3: 19F NMR yields or isolated yields are shown?

It is an isolated yield. In the experimental section, we have described in detail.

Lines 116 & 117: something is missing in “( = 0)”

It would be a corrupted letter (delta).

Pages 5 & 6: symbols 3a–3g should be bolded.

It would be a corrupted letter. 

Round 2

Reviewer 1 Report

The use of CuI is a very convenient method compared to the pervious reports and I believe it will be very interesting to the scientific community.  As I mentioned in the previous review, the introduction was written very general. It should be written with a little bit more depth and specificity to be more appealing to the scientific community. For example, what are the advantages of this work compared to the similar articles for example compared to the references 31 and 33? Why this work is significant? etc..

Author Response

The use of CuI is a very convenient method compared to the pervious reports and I

believe it will be very interesting to the scientific community. As I mentioned in the

previous review, the introduction was written very general. It should be written with a

little bit more depth and specificity to be more appealing to the scientific community.

For example, what are the advantages of this work compared to the similar articles for

example compared to the references 31 and 33? Why this work is significant? etc..

Thank you for the useful suggestion. Compared to the previous reports, our

protocol has advantages to use CuI (inexpensive catalyst) and Ar-I (I; a

general leaving group). We have added a sentence into the conclusion part.

Using a simple combination of the coupling partners (iodoarenes and Me3Si-

CF2PO(OEt)2), ... under mild reaction conditions. The present transformations

employng CuI are synthetically useful.

Reviewer 2 Report

The ms has been improved, however, it still suffers certain limitations.

-         “ even now, there have been many publications on Cu-mediated aromatic fluoroalkylation which need the use of a stoichiometric amount of copper salts. In this paper, we would like to disclose a new entry of Cu-catalyzed difluoromethylation.”

In my opinion, catalytic method should not be emphasized, as the yields were low in these cases. It is mentioned even in the Conclusions as the major novelty, however NMR yields are below 50%, meaning that preparative yields are even lower, while stoichiometric methods result in much higher yields. Scheme 3 also shows that the catalytic amount of CuI is not enough

-        “Furthermore, in Table 2, we have added two examples (3h and 3i) with chloro- and bromo-functionalities.”

Are these compounds new? If so, these should be fully characterized!

-        The major role of CsF is the activation of Me3Si-CF2PO(OEt)to generate CF2PO(OEt)anion. However, the real role (the behavior in THF solvent) is not clear due to the low solubility of CsF in organic solvents such as ethers.

This could be mentioned in the ms.

-        The major by-product was HCF2PO(OEt)2. For instance, the reaction of Py-I (1f) with 2 gave 25% of HCF2PO(OEt)2 besides the desired Py-CF2P(O)(OEt)2 (3f).

This could be mentioned in the ms.

-        “Before using Me3SiCF2P(O)(OEt)2, we conducted many experiments using BrZnCF2PO(OEt)2 in the presence of a small amount of Cu salts. However, TON of the Cu-catalyzed cross-coupling of simple aryl iodides was low (TON: 1.0-1.2). The present studies (the use of Me3SiCF2P(O)(OEt)2) would provide a synthetic potential of Cu- catalyzed transformations.”

How about TON in the latter case using Me3SiCF2P(O)(OEt)2?

-  English usage is poor in some cases. Only the listed errors were corrected, however, a general correction and proofreading is needed. See eg. line 121 “…would be render a reaction catalytic in copper possible.”  or line 171: “A Following a general..”, etc.

-        We have removed the corresponding sentences, moves into “the former part of 3. Materials and Methods” and changed as follows.

NMR data should be assigned and systematically compared with that of the literature data instead of saying "spectra matched that reported by XY". This is still an issue.

A general procedure should be written for each compounds instead of copy-pasting the same procedure 8 times.

If 3h and 3i are new compounds, characterization is missing. If this is not the case, new coumpounds should be added to extend the scope and limitation and increase novelty.

In conclusion, the ms still needs a major revision.

Author Response

In my opinion, catalytic method should not be emphasized, as the yields were low in

these cases. It is mentioned even in the Conclusions as the major novelty, however NMR

yields are below 50%, meaning that preparative yields are even lower, while

stoichiometric methods result in much higher yields. Scheme 3 also shows that the

catalytic amount of CuI is not enough.

Thank you for your comments. In the Cu-mediated methods, the use of a stoichiometric

amount of copper salts is general. We believe that some (reported) Cu-mediated

reactions would proceed by small amount of copper salts. In this paper, although the

TON was not high, we want to show the possibility to find the catalytic

difluoromethylenephosphonation.

Are these compounds new? If so, these should be fully characterized!

Both products 3h and 3i are known compounds.

(3h: CAS No. 2155816-80-9; 3i: CAS No. 177284-56-9)

This could be mentioned in the ms.

- The major role of CsF is the activation of Me3Si-CF2PO(OEt)2 to generate

CF2PO(OEt)2 anion. However, the real role (the behavior in THF solvent) is not clear

due to the low solubility of CsF in organic solvents such as ethers.

We have added the following sentence into page 4, the bottom of the paragraph.

The major role of CsF is the activation of organosilicon compound 2 to generate

CF2PO(OEt)2 anion. The low solubility of CsF in THF would contribute to slow

generation of CF2PO(OEt)2 anion and formation of CuCF2PO(OEt)2 species in

the catalytic reactions.

This could be mentioned in the ms.

- The major by-product was HCF2PO(OEt)2. For instance, the

reaction of Py-I (1f) with 2 gave 25% of HCF2PO(OEt)2 besides the

desired Py-CF2P(O)(OEt)2 (3f).

We have added the following sentence into pages 3-4, in the description of the reactions

using a stoichiometric amount of CuI.

In the each cross-coupling of 1 with 2, the major by-product was HCF2PO(OEt)2;

for instance, the reaction of 2-iodopyridine (1f) with 2 gave 25% of HCF2PO(OEt)2

besides the desired Py-CF2P(O)(OEt) 2 (3f).

How about TON in the latter case using Me3SiCF2P(O)(OEt)2?

- “Before using Me3SiCF2P(O)(OEt)2, we conducted many experiments using

BrZnCF2PO(OEt)2 in the presence of a small amount of Cu salts. However, TON of the

Cu-catalyzed cross-coupling of simple aryl iodides was low (TON: 1.0-1.2). The present

studies (the use of Me3SiCF2P(O)(OEt)2) would provide a synthetic potential of Cucatalyzed

transformations.”

The best TON of the present cross-coupling was about 7 (for the compound 3g,, one of

the activated aryl iodides). The values of TON depend on the structure of the substrates,

we think.

Reviewer 3 Report

In the revised version of the manuscript authors added new experimental results and new text, including the Poisson's reaction in Scheme 2. Unexpectedly, with the new data there are more doubts on the paper.

First of all, it is obvious now that in the described method stoichiometric amounts of CuI are required and the yields are clearly dependent on the amount of CuI present in the reaction mixture. On the other hand, it seems obvious that the authors intended to develop a CuI-catalyzed approach. Unfortunately, they failed. It happens but it should be clearly written. The only successful use of CuI in catalytic amounts was found for reaction of 2-iodoquinoline. Surprisingly, it does not seem that authors payed too much attention to this exceptional case.

Thus, other advantages of the presented approach should be emphasized. Better yields? Higher rates? More economic? Simplicity? Possibility to prepare compounds that were inaccessible by other methods? At the moment I can see only a combination of known numerous reactions of type Ar-I + X-CF2P(O)(OEt)2 -> Ar-CF2P(O)(OEt)2 and of type Ar-Z + TMS-CF2P(O)(OEt)2 -> Ar-CF2P(O)(OEt)2.

Apart from the necessity to reorganize the presentation of the experimental material some other issues were found now:

1.      A conclusion "In  some cases,  a  small  amount  of  CuI  promoted  the  cross-coupling  reactions  to  afford aryl(difluoromethyl)phosphonates" is poorly supported by the results presented. The same applies to "(or  a  catalytic  amount)" in Abstract.

2.      Some yields are given as judged from 19F NMR, some are isolated yields. Compound 3g was apparently included in basic studies and 19F NMR yields should be given to allow comparison with other results. Similarly, a 19F NMR yield of 3f using 1 eq CuI is lacking.

3.      The authors stated (lines 119-212): "Control of  the  slow  generation  of  CF2PO(OEt)2   anion  [note by the referee: anions are ions, the minus sign is lacking] by  the  reaction  of  Me3Si-CF2PO(OEt)2   (2)  with  CsF would be render a reaction catalytic in copper possible". Is it a conclusion (however, I cannot see a support for this in the results) or a justification for starting the reported studies (then, however, it should be in the Introduction)? More detailed discussion on this topic is required to make the authors' idea clear. Additionally, grammar of this sentence must be corrected.

4.      Footnotes "1" to table 1 (time) and table 2 (time & activators) are not consistent with the data in reaction equations.

Concluding, the manuscript requires significant corrections in presentation of the material, with clear presentation of the current state of art, the authors' ideas what may be improved and how it can be done, and the results – both successes and failures.

Author Response

In the revised version of the manuscript authors added new experimental results and

new text, including the Poisson's reaction in Scheme 2.

Unexpectedly, with the new data there are more doubts on the paper. First of all, it is

obvious now that in the described method stoichiometric amounts of CuI are required

and the yields are clearly dependent on the amount of CuI present in the reaction

mixture. On the other hand, it seems obvious that the authors intended to develop a CuIcatalyzed

approach. Unfortunately, they failed. It happens but it should be clearly

written. The only successful use of CuI in catalytic amounts was found for reaction of 2-

iodoquinoline. Surprisingly, it does not seem that authors payed too much attention to

this exceptional case.

Thus, other advantages of the presented approach should be emphasized.

Better yields? Higher rates? More economic? Simplicity? Possibility to prepare

compounds that were inaccessible by other methods? At the moment I can see only a

combination of known numerous reactions of type Ar-I + X-CF2P(O)(OEt)2 -> Ar-

CF2P(O)(OEt)2 and of type Ar-Z + TMS-CF2P(O)(OEt)2 -> Ar- CF2P(O)(OEt)2.

Thank you for the comments. Compared to the previous reports, our

protocol has advantages to use CuI (inexpensive catalyst) and Ar-I (I; a

general leaving group). We have added a sentence into the conclusion part.

Using a simple combination of the coupling partners (iodoarenes and Me3Si-

CF2PO(OEt)2), ... under mild reaction conditions. The present transformations

employng CuI are synthetically useful.

1. A conclusion "In some cases, a small amount of CuI promoted the cross-coupling

reactions to afford aryl(difluoromethyl)phosphonates" is poorly supported by the results

presented. The same applies to "(or a catalytic amount)" in Abstract.

The expression “in some cases” reflects our results. As pointed out, this is the weak

point, particularly in the present reactions catalytic in copper.

We have cahnged the sentence in the abstract as follows.

Upon treatment with a stoichiometric amount (or a catalytic) of CuI and CsF,…

-> Upon treatment with a stoichiometric amount (or a catalytic amount in some

cases) of CuI and CsF,…

2. Some yields are given as judged from 19F NMR, some are isolated yields. Compound

3g was apparently included in basic studies and 19F NMR yields should be given to

allow comparison with other results. Similarly, a 19F NMR yield of 3f using 1 eq CuI is

lacking.

After the first revision, in Table 2, we changed to the yield of 3 to isolated yields. In the

cases of phosphonates, the yields estimated by 19F NMR analyses are not so largely

different from those of isolated. However, for compound 3d (OMe), there is the problem

to isolate 3d from the by-product HCF2PO(OEt)2, therefore we described 19F NMR

yield of 3d in Table 2.

3. The authors stated (lines 119-212): "Control of the slow generation of CF2PO(OEt)2

anion [note by the referee: anions are ions, the minus sign is lacking] by the reaction of

Me3Si- CF2PO(OEt)2 (2) with CsF would be render a reaction catalytic in copper

possible". Is it a conclusion (however, I cannot see a support for this in the results) or a

justification for starting the reported studies (then, however, it should be in the

Introduction)? More detailed discussion on this topic is required to make the authors'

idea clear. Additionally, grammar of this sentence must be corrected

Following to the suggestion, we have changed the sentences in the bottom of the

paragraph.

Control of the rate of the generation of CF2PO(OEt)2 anion by the reaction of

Me3Si-CF2PO(OEt)2 (2) with CsF would render a reaction catalytic in copper

possible. The major role of CsF is the activation of organosilicon compound 2 to

generate CF2PO(OEt)2 anion. The low solubility of CsF in THF would contribute

to slow generation of CF2PO(OEt)2 anion and formation of CuCF2PO(OEt)2

species in the catalytic reactions.

4. Footnotes "1" to table 1 (time) and table 2 (time & activators) are not consistent with

the data in reaction equations.

In the footnote, we have corrected the reaction time (24 h, Tables 1&2) and activator

(CsF, , Table 2).